# Costing the Scale-Up of a National Primary School-Based Fluoride Varnish Program for Aboriginal Children Using Dental Assistants in Australia

**DOI:** 10.3390/ijerph17238774

**Published:** 2020-11-26

**Authors:** John Skinner, Yvonne Dimitropoulos, Boe Rambaldini, Thomas Calma, Kate Raymond, Rahila Ummer-Christian, Neil Orr, Kylie Gwynne

**Affiliations:** 1Poche Centre for Indigenous Health, Room 224 Edward Ford Building, The University of Sydney, Sydney 2006, Australia; yvonne.dimitropoulos@sydney.edu.au (Y.D.); boe.rambaldini@sydney.edu.au (B.R.); tom.calma@sydney.edu.au (T.C.); neil.orr@mq.edu.au (N.O.); kylie.gwynne@mq.edu.au (K.G.); 2Department of Health, Northern Territory Government Level 4, Darwin 0800, Australia; kate.raymond@nt.gov.au; 3Fluoride Varnish Initiative, Loddon Mallee Aboriginal Reference Group, Bendigo 3550, Australia; Rahila.Christian@bdac.com.au; 4Faculty of Medicine, Health and Human Sciences, Macquarie University, Sydney 2113, Australia

**Keywords:** Fluoride varnish, dental assistants, Aboriginal, oral health, implementation science, scale-up

## Abstract

There is good evidence that fluoride varnish programs are effective in preventing dental caries in children. This study aims to provide a costing for the scale-up of a child fluoride varnish program in New South Wales (NSW), Australia. Most child fluoride varnish programs are school-based, and a number of studies have examined the acceptability and cost effectiveness of using non-dental providers to apply the fluoride varnish. This paper describes the number of primary schools in Australia that could be targeted using a standard population-based risk criteria based on published data. A costing method was developed for various scenarios of school enrolment and provider types, along with potential revenue from the Child Dental Benefits Schedule (CDBS). Most of the costs of a school-based fluoride varnish program can be covered by the CDBS with assumptions of 80% child consent and 75% CDBS eligibility. While the scale-up of child fluoride varnish programs to prevent dental caries has been recommended by numerous strategic plans and reports, particularly for Aboriginal and Torres Strait Islander children, limited progress has been made. This paper concludes that using a standardized criteria for targeting schools using a combination of ICSEA and Aboriginal enrolments, and aiming at four applications a year, is feasible, and that the main costs of the program could be covered by using the CDBS.

## 1. Introduction

There is good evidence that fluoride varnish programs are effective in preventing dental caries in both deciduous and permanent teeth if applied two or more times per year [1,2,3]. These programs are particularly important amongst high risk populations, including those without access to water fluoridation and Aboriginal and Torres Strait Islander communities [4,5,6]. Even in areas with access to water fluoridation, fluoride varnish programs are effective in preventing dental disease in low socio-economic groups and are complementary to other programs, such as tooth brushing and toothpaste [2]. Preschool and school-based fluoride varnish programs are in place in many countries, including Canada, the United States, Sweden, and Scotland [7,8,9,10]. In Australia, the largest programs are in the Northern Territory, Western Australia, and more recently Tasmania [11,12,13]. These programs are an important part of the prevention spectrum between the whole of population oral health approaches of water fluoridation and smaller scale tooth brushing and toothpaste programs, and the regular dental visits to dental and oral health professionals [7,8,9,10,11,12,13].

The greater use of fluoride varnish programs in Australia is a major strategy of the National Oral Health Plan [6]. The National Oral Health Plan noted the important role that fluoride varnish programs can play in high-risk communities, fluoridated or unfluoridated, to increase the availability of fluoride regularly to children who may not have regular access to dental care and/or fluoride toothpaste at home [6]. The Aboriginal Oral Health Plan provides further support for the scale-up of a fluoride varnish program in New South Wales (NSW) in communities with high Aboriginal populations and/or with no access to water fluoridation [14]. These plans support the implementation and scale-up of fluoride varnish programs both at the state and national level in Australia [6,14]. In Australia, there are state, national, and professional association guidelines that also support the use of fluoride varnishes in high risk communities [3,5,15,16]. These guidelines reinforce the need to see a greater implementation of these programs in keeping with Australia’s National Oral Health Plan.

The National Oral Health Plan also recognizes the important role fluoride varnish can play in the prevention of caries and the need to utilize a non-dental workforce to apply fluoride varnish, and specifically mentioned the need to overcome legal and policy barriers for dental assistants to apply fluoride varnish [5]. Various national and state plans, guidelines, and policies, along with the Australian Medical Association (AMA) Report Card on Indigenous Health, all acknowledge the importance of fluoride varnish, as well as the use of an expanded non-dental professional workforce to apply it [3,15,16,17]. Several specifically mention the need to use dental assistants in scaled-up programs, especially for Aboriginal children [6,14,15,17].

While all national and NSW state oral health planning documents recommend the expansion of fluoride varnish programs, limited progress has been made nationally, with the notable exceptions of the Northern Territory, Western Australia, and Tasmania [11,12,13]. This study aims to provide a costing for the scale-up of a child fluoride varnish program in New South Wales (NSW), Australia by modeling the costs of scaling up a school-based fluoride varnish program that has been piloted in Indigenous communities in NSW [18,19,20].

## 2. Materials and Methods

This paper reported on a descriptive study that estimated the number of primary schools that need to be targeted by a national implementation of a school-based fluoride varnish program for disadvantaged schools with high enrolments of Indigenous students. The estimates are based on schools having a proportion of Aboriginal and Torres Strait Islander students of 12% or greater, and an Index of Community Socio-Educational Advantage (ICSEA) rating of 1050 or less. ICSEA is a composite scale that represents levels of educational advantage, where lower ICSEA values indicate lower levels of educational advantage [21]. The index was developed by Australian Curriculum, Assessment and Reporting Authority (ACARA), and is used to compare educational advantage across public, Catholic, and independent schools [21]. The ICSEA ranking data, along with data on Indigenous enrolments, was obtained from the ACARA web site from the School Profiles for 2017 [22]. This allowed the creation of a national list of schools by state and territory that met the conditions of an ICSEA ranking of 1050 or less and a proportion of Aboriginal and Torres Strait Islander students enrolled of 12% or more [20] from a public national data set [22].

The direct costs of a school-based fluoride varnish program (FVP) relating to several scenarios are described using estimates and administrative data from a range of sources [19,20,23]. The main components of the costing were the fluoride varnish, labor, travel, and consumables. These key components were based on previous costing work undertaken in Sweden, Ireland, and the United States, along with pilot studies in NSW undertaken by the Poche Centre for Indigenous Health in partnership with Aboriginal Community Controlled Health Services and Local Health Districts (LHDs) [10,18,19,20,24,25].

The standard fluoride varnish day consent processes and application procedures are outlined elsewhere [18,19,20]. While a number of programs include oral health education, water bottles, fissure sealants, and/or tooth brushing, we wanted to specifically cost the standard fluoride varnish component in keeping with the scale-up goals of various state and national plans [6,14].

We costed the labor component of the estimates based on two scenarios: firstly, a dental assistant applying the fluoride varnish, and secondly, an oral health therapist undertaking this work. While the pay scales for these roles vary from state to state and by qualification and experience level, we used salary levels indicative of our experience to date in implementing pilot programs in NSW [18,19,20]. In our calculations, we assumed that each of these workers could apply fluoride varnish to up to 80 children per day based on our previous pilots and standard protocol [18,19,20]. This is based on an estimate of three minutes per fluoride varnish application per child. Realistically, there is only about four and a half hours of access time to students in a typical school day once allowance is made for recess and lunch, as well as setup.

The main material costs were fluoride varnish (Duraphat) [26] and consumables, such as face masks, gloves, gauze, and anti-microbial hand gel (Duraphat is a brand of fluoride varnish commonly used in Australia and is manufactured by Colgate Palmolive [26]). Duraphat comes in a 10 mL tube that can make 25 applications (0.4 ml per application) and costs AUD$55.10 per tube.

The utilization of the CDBS represents a revenue stream for state and territory health systems, but could also be used to offset the costs of the fluoride varnish program scale-up. For each eligible individual, CDBS item (88121) can only be claimed once in a six-month period, and currently attracts a benefit of $34.55 [27]. Thus, currently only two applications of fluoride varnish can be claimed from the CDBS in a calendar year, with the maximum amount that could be claimed per eligible child being $69.10.

We included in our modelling variations in CDBS eligibility. While most Aboriginal children in high-risk communities are likely to be CDBS eligible [28], we allowed for two scenarios: 80% and 90% eligibility. However, based on the historically low uptake of the CDBS by Aboriginal families, the majority of families are likely to have sufficient value remaining under their $1,000 cap for the claims for the fluoride varnish items to be successful [28]. This modelling was applied overall to all NSW schools that met the criteria above as a scale-up case study. We also assessed the readiness for a state and national scale-up process using the NSW Ministry of Health Guidelines on increasing the scale of population health interventions and by using and developing a program logic model [29,30] to reflect the status of scale-up planning in Australia (Appendix A).

## 3. Results

### 3.1. National Results

Using the combined criteria of ICSEA of 1050 or less and Indigenous enrolments of 12% or more, 1646 primary and combined primary/secondary schools were identified nationally [13,20]. Table 1 presents these data by state and territory, with NSW having the highest number of schools (*n* = 619), followed by QLD (*n* = 411) and WA (*n* = 212).

### 3.2. New South Wales Case Study

The number of schools in NSW by LHDs meeting our selection criteria [14,21] for the scaled up national fluoride varnish program is outlined in Table 2. The greatest number of schools is in the Hunter New England LHD, and this LHD also has the highest level of enrolment. This is followed by Western NSW, Murrumbidgee, and North Coast LHDs. These smaller LHDs will require additional resources for program administration if they are to implement the scaled-up program.

Table 3 outlines the main material costs of a fluoride varnish day, including the fluoride varnish itself, personal protective equipment, anti-microbial hand gel, gauze, micro-applicators, and plastic bags for rubbish. Approximate total material costs are presented based on different numbers of students seen per day.

Table 4 outlines the costs of a fluoride varnish day in terms of labor and material costs. These include modelling based on different average class sizes, levels of consent, and number of students seen per day. We assumed that in some cases there may be a single person undertaking the program in small schools, so we applied extra setup and pack-up time for enrolments of 100 or more, which meant they could be done over more than one day.

Table 5 presents various revenue models using CDBS under two different assumptions. Model 1 assumes a parent/guardian consent rate of 80% or more and that the child’s CDBS has a sufficient balance to pay for both items being claimed. Model 2 assumes a parent/guardian consent rate of 80% or more and that the child’s CDBS has a sufficient balance to pay for at least 75% of the value of the items being claimed.

## 4. Discussion

Several papers and guidelines have evaluated the costs related to the implementation of fluoride varnish programs [24,25] or cost effectiveness of fluoride varnish programs comparted with fissure sealants [32,33,34]. A number of these evaluations included costing information. The costing of a fluoride varnish program in Ireland used a similar methodology to the one we have developed for this paper [18]. The main difference between our paper and theirs was the assumption around average time per fluoride varnish application, and we did not include costing for portable stools, chairs, or lights, as we assumed that these would be available at the settings. We have assumed an average fluoride varnish application per student of three minutes, whereas the study from Ireland used six minutes [18]. However, a review of fluoride varnish programs by Mishra et al. found a range of one to four minutes per application for preschool children, depending on the number of teeth present [33]. Our timing based on our protocol and recent pilot study [18,19,20] suggests that the three-minute estimate is reasonable. The study from Ireland also found that the main material cost was the fluoride varnish, like we did [24]. In addition, Neidell et al. found that the use of school health aides rather than dentists or mid-level dental providers helped to reduce costs and improve the cost effectiveness of fluoride varnish over sealants [33]. This concurs with our modelling, finding improved cost effectiveness of dental assistants over oral health therapists.

Our model also looks at the CDBS as a potential revenue stream for state-based services. The current CDBS funding rules allow for two fluoride varnish applications per child to be claimed in a calendar year if they are CDBS eligible [27]. Given that the schools being targeted in the proposed scale-up are in low SES areas and have high Aboriginal enrolments, we assumed that between 75–100% of students would be eligible. We estimate that there would be a loss of revenue of 5–10% due to procedural, eligibility, or cap issues. Therefore, we modelled both at the 100% and 75% successful claim levels. Recent consent rates in two NSW studies suggest that the 80% consent assumption is likely to be correct in terms of the number of children participating [19,20], but data on CDBS eligibility was not collected. The additional revenue from larger schools can also be used to offset the additional costs of the smaller schools.

The Tasmanian Fluoride Varnish and Sealant Program uses the ICSEA scale to target schools, and we have also used this in our methodology [13]. We have found that using an ICSEA of 1050 or less and an Aboriginal enrolment of 12% or more produced a 100% fit with primary schools already in our pilot programs. The ICSEA was also one of the measures used to describe the socioeconomic aspects of child oral health in the report on the results of the National Child Oral Health Survey 2012–14 [35]. The highest prevalence of dental caries experience for Australian children was found on those children in ICSEA1 (< 986) followed by ICESEA2 (986–1044) [35]. More recently, the NSW Ministry of Health has used ICSEA to target schools for its new Child Mobile Dental Van Program that also includes fluoride varnish applications [36].

The number of fluoride varnish applications each year as part of a program greatly influences the overall cost of the program. The evidence suggests that two or more applications provide the best preventive benefit of fluoride varnish [1,2,3]. In our scale-up planning, we have designed a once per term (four applications a year) program. This aims to apply fluoride varnish to children at least three times per year, thus allowing for students to be occasionally absent on scheduled fluoride varnish days. A recent report on the Northern Territory Remote Aboriginal Investment: Oral Health Program found that in 2017, 44% of girls and 41% of boys only received one fluoride varnish application [13]. A similar proportion of boys (41%) and girls (46%) received at least two fluoride varnish applications. Two recent studies in NSW primary schools found that aiming at four applications per year was acceptable to parents and schools, and that most children had at least two applications in a year [19,20].

During the development of this costing paper, the fourth review of the Dental Benefits Act that is used by the CDBS was undertaken by the Commonwealth [28]. One of the authors wrote on behalf of the Poche Centre for Indigenous Health to the Chair of the Review Committee requesting that the group considers allowing up to three fluoride varnish applications per year to be claimed under the CDBS if they were a part of a structured fluoride varnish program. The submission was considered by the Committee, but not supported [29]. Nevertheless, the Committee noted the need to establish a formal process to consider future variations to the Dental Benefits Act 2008 [28]. It is important to note that the Australian Dental Association has recently updated its Use of Fluorides Guidelines, and these address the issues of need, safety, and clinical effectiveness, along with the use of dental assistants [15], that would support national scale-up and greater use of the CDBS.

Several Australian states have recently commenced the introduction or re-introduction of child dental services in schools, often using mobile dental vans. These services, while often popular politically and with some schools and parents, have not been shown to be cost effective [37,38], particularly given the wider population impact of water fluoridation in many Australian states, along with the greater use of fluoride toothpaste in lowering overall dental disease levels. A scaled-up fluoride varnish program with rebates claimed via the CDBS is likely to be a more cost-effective option. In addition, with the low uptake of the CDBS by Aboriginal children [29], this would be an important preventive adjunct to the existing scheme. This may also raise the number of dental visits by Aboriginal children under the CDBS.

The study has a number of limitations, including the fact that the assumption of at least 80% parental consent (as was achieved in the pilot projects) may not be achievable in a program that is run across the state. Furthermore, the costings may not be readily generalizable because of differences in the classification, labor costs, and potential differences in quality of fluoride varnish application between oral health therapists and dental assistants. Indirect costs were not calculated, as the data to calculate these was available to us, but we acknowledge that these costs could be substantial in some cases.

## 5. Conclusions

This paper concludes that using a standardized criteria for targeting schools using a combination of ICSEA and Aboriginal enrolments, and aiming at four applications a year, is feasible, and that the main costs of the program could be covered by using the CDBS even when only two applications can be claimed.

## Figures and Tables

**Table 1 ijerph-17-08774-t001:** All primary/combined schools where ICSEA is less than 1050 and/or Aboriginal enrolments are 12% or more, by state and territory, 2017 ^1^.

State/Territory	Total Schools Meeting Criteria	Total Student Enrolment	Average Enrolment (sd)	Major Cities	Inner and Outer Regional	Remote and Very Remote
NSW	619	121,024	195.5 (179.0)	143	433	43
Vic	81	9974	123.1 (141.2)	5	75	1
SA	98	184,50	188.3 (150.5)	35	40	23
WA	212	43,580	205.6 (182.0)	47	57	108
NT	139	21,890	157.5 (148.8)	0	29	110
TAS	81	18,831	232.5 (157.0)	0	74	7
Qld	411	101,840	247.8 (277.0)	47	269	95
ACT	5	973	194.6 (241.5)	4	1	0
Australia	1646	336,562	204.5 (205.0)	281	978	387

^1^ Includes government, Catholic, and independent schools. Data Source: Reference [22].

**Table 2 ijerph-17-08774-t002:** All New South Wales primary schools where ICSEA is less than 1050 and/or Aboriginal enrolments are 12% or more, by Local Health District, 2017 ^2^.

Local Health Districts	Total Schools Meeting Criteria	Total Student Enrolment	Average Enrolment (sd)
Central Coast	28	9030	322.5 (197.2)
Far West	13	1725	132.7 (103.2)
Hunter New England	173	34,672	200.4 (197.4)
Illawarra Shoalhaven	26	6399	246.1 (131.1)
Mid North Coast	49	9436	192.6 (160.1)
Murrumbidgee	64	9811	153.3 (127.3)
Nepean Blue Mountains	20	3277	163.9 (121.2)
Northern NSW	58	10,772	185.7 (213.7)
Northern Sydney	3	262	87.3 (98.6)
South Eastern Sydney	5	814	162.8 (139.4)
South Western Sydney	31	9201	296.8 (154.2)
Southern NSW	22	3835	174.3 (166.3)
Sydney	5	1327	265.4 (270.3)
Western NSW	111	17,271	155.6 (165.7)
Western Sydney	11	3192	290.2 (163.4)
NSW	619	121,024	195.5 (179.0)

^2^ Includes government, Catholic, and independent schools. Data Source: Reference [22].

**Table 3 ijerph-17-08774-t003:** Estimated costs of materials for the school-based fluoride varnish program.

Materials	Unit Cost	Quantity/Day	Costing at 25 Students	Costing at 50 Students	Costing at 100 Students
Fluoride Varnish	$55.10	6	$55.10	$110.20	$220.40
Gloves (box of 200)	$19.40	2	$4.85	$9.70	$19.40
Masks (box of 50)	$15.73	3	$7.87	$15.73	$31.46
Antimicrobial hand gel (500mL bottle) ^1^	$28.87	1	$4.51	$9.02	$18.04
Gauze (packet of 100)	$3.49	2	$1.74	$3.49	$6.98
Micro-applicators (box of 400)	$17.38	1	$1.09	$2.17	$4.35
Plastic bags (20 rolls of 50)	$55.30	1	$0.05	$0.05	$0.05
Total			$75.21	$150.36	$300.68

^1^ Based on the 3 mL recommendation for each session of hand hygiene by the WHO [31].

**Table 4 ijerph-17-08774-t004:** Cost models for a support a school-based fluoride varnish program based on different enrolment levels and provider type.

Enrolment	Consent (80%)	Labor DA	Labor OHT	Consumables ^1^	Travel ^2^ DA (Regional)	Travel ^2^ OHT (Regional)	Total Costs DA (Regional)	Total Costs OHT (Regional)
25	20	$288	$371	$241	$86	$111	$615	$723
50	40	$384	$495	$481	$115	$148	$980	$1125
100	80	$576	$743	$962	$173	$223	$1711	$1928
150	120	$960	$1238	$1443	$288	$372	$2691	$3053
200	160	$1152	$1486	$1924	$346	$446	$3422	$3856
250	200	$1344	$1734	$2406	$403	$520	$4153	$4659
300	240	$1536	$1981	$2887	$461	$594	$4884	$5463

^1^ Consumable costs includes the fluoride varnish. ^2^ Two hours of labor per day is costed for setup and pack-up. Regional travel and subsistence costs were included at 30% of total labor costs.

**Table 5 ijerph-17-08774-t005:** Revenue modelling to support a school-based fluoride varnish program.

Enrolment	Consent (80%) ^1^	Costs DA (Regional)	Costs OHT (Regional)	Total CDBS Revenue Model 1 (100%) ^2^	Total CDBS Revenue Model 2 (75%)
25	20	$615	$723	$1382	$1037
50	40	$980	$1,125	$2764	$2073
100	80	$1,711	$1,928	$5528	$4146
150	120	$2,691	$3,053	$8292	$6219
200	160	$3,422	$3,856	$11,056	$8292
250	200	$4,153	$4,659	$13,820	$10,365
300	240	$4,884	$5,463	$16,584	$12,438

^1^ Consent assumed at 80% on average; ^2^ CDBS eligibility is modelled at 100% and 75%.

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
