# Peer review of "Costing the Scale-Up of a National Primary School-Based Fluoride Varnish Program for Aboriginal Children Using Dental Assistants in Australia"

_ijerph, 2020, doi:10.3390/ijerph17238774_

Round 1

Reviewer 1 Report

Points for consideration:

Line 28: change precent to prevent

Line 28. The words starting with While are not a sentence. Consider adding after dental caries "has been recommended by numerous strategic plans and reports"

Lines 8, 135 and 146: Use upper case C in catholic

Author Response

Point 1: Line 28 change precent to prevent

Response 1: Typographical error now corrected in amended manuscript (see Line 28).

Point 2: Line 28. The words starting with While are not a sentence. Consider adding after dental caries "has been recommended by numerous strategic plans and reports"

Response 2: Suggested amendment made in revised manuscript in Line 28.

Point 3: Lines 8, 135 and 146 Use upper case C in catholic

Response 3: Catholic now capitalised in amended manuscript (see Lines 83, 136, and 148).

Reviewer 2 Report

None

Author Response

Point 1: Please clarify the type of the article, eg Research?, etc....

Response 1: Research Article. This has been amended in Line 1.

Point 2: Statistical model?

Response 2: No, it’s not a statistical model. It’s a comparison of actual costs under different scenarios. We have clarified that we are calculating whether revenue from a Commonwealth government funded dental program – the Child Dental Benefits Schedule can meet the main costs of a school-based fluoride varnish program under several scenarios. We have amended this sentence to remove “estimates” and “comparison” so it is less suggestive of a statistical model (see Line 25)

Point 3: Is it a Conclusion?

Response 3: We have clarified that this is a conclusion and have amended this sentence to read “This paper concludes that….” (see Line 30)

Point 4: reference(s)?

Response 4: The references have been added as requested in Line 43.

Point 5: reference(s)?

Response 5: The references have been added as requested in Line 45.

Point 6: Avoid using parenthesis....

Response 6: The parenthesis have been removed (see Line 51).

Point 7: Reference(s)?

Response 7: The references for this sentence have been added (see Line 56).

Point 8: Write clearly the aim of the study

Response 8: The aims of the study have been added to the Introduction (see Lines 71-74).

Point 9: The Authors must clarify if a statistical model was

Response 9: The study is descriptive, rather than a statistical model. This has been clarified in the amended manuscript (see Line 77).

Point 10: Please express in %

Response 10: Per cent changed to % in the amended manuscript (see Line 80).

Point 12: reference(s)?

Response 12: The references have been added as requested (see Line 89).

Point 13: Reduce Para

Response 13: The paragraph has been reduced in the amended manuscript. (see Lines 90-91).

Point 14: reference(s)?

Response 14: The references have been added as requested in the amended manuscript (see Line 91).

Point 15: reference(s)?

Response 15: The references have been added as requested in the amended manuscript (see Line 106).

Point 16: Remove this phrase

Response 16: The phrase has been removed as requested and the cost of the Duraphat included in the previous sentence (see Lines 112-113).

Point 17: Irrelevant data without interesting for the median…

Response 17: Sentence amended in manuscript to make it more relevant to the paper (see Line 114-116).

Point 18: reference(s)?

Response 18: A reference has been added as requested (See Line 123).

Point 19: Remove this phrase

Response 19: Phrase removed with part of it now included in the next sentence (see Lines 124-126).

Point 20: reference(s)?

Response 20: The references have been added as requested in Line 135.

Point 21: Inclusion/Exclusion criteria? How did the authors determine the study sample? Protocol? References(s)? Possible selection and recall biases …….….

Response 21: The references have been added as requested to cover Points 20 and 22 and these provide the relevant references and protocol. We have also added the public data source that we used to develop the table by applying our protocol (see Lines 142 and 154).

Point 22: Where are? reference(s)?

Response 22: The references have been added as requested in the amended manuscript (see Line 145).

Point 23: What are the meaning of these symbols?

Response 23: Mean and standard deviation (sd).  “(sd)” added to Tables 1 and 2 in the amended manuscript.

Point 24: reference(s)?)

Response 24: The references have been added as requested in Line 162.

Point 25: 170? 171?

Response 25: Typos already corrected in amended manuscript by Editorial Team.

Point 26: By whom?

Response 25: The sentence has been updated to refer to the CDBS funding rules and a reference has also been added in the amended manuscript (see Lines 211-212).

Point 27: Please do not repeat results

Response 27: Two sentences that repeated results have been deleted in the amended manuscript (see Line 217-218).

Point 28: (JS)?

Response 28: Deleted in amended manuscript (see line 241).

Point 29: Please use a more scientific style of

Response 29: Direct quote removed in the amended manuscript (see Line 244).

Point 30: See previous remark

Response 30: Direct quote removed in the amended manuscript (see Line 245).

Point 31: What about the limitations of the study

Response 31: A limitations section has been added in the amended manuscript (see Lines 259-265).

Point 32: Reduce this section and mention the main conclusion(s) only.

Response 32: Conclusion reduced in the amended manuscript (see Line 266).

Point 33: Correct the style of pages presentation

Response 33: Corrected in the amended manuscript (see Line 312).

Reviewer 3 Report

Caries is a public health problem that continues to affect babies and school children worldwide. Fluoride varnish which is one of the most important materials to prevent tooth decayed is easy to apply and well tolerated by children. Fluoride varnish program using dental assistants is a positive and meaningful school-based program and I think that it will be a great benefit to the aboriginal children in Australia.

However, the significance of prevention lies in the development of good tooth-brushing behavior in daily life. The weakness of this program background does not explain the current tooth-brushing behavior among of children, and the background, measures, and related results of domestic fluoride applications in Australian.

Materials and Methods

This article lacks the procedures for applying fluoride in the Materials and Methods section. Should fluoride be applied after teeth cleaning? Do you implement relevant oral hygiene education during the fluoride application process? It is recommended that the author should talk more about the oral hygiene status about these issues.

Result

Cost of Labour DA ($403) was much higher than Labour OHT ($206). Please check it $403 of Labour DA column in Table 4.

Discussion

Fluoride varnish is painted on the top and sides of each tooth with a small brush. A small brush is The small brush is a consumable product, and the small brush is not mentioned in the material and cost estimation.

All health policies need to evaluate costs and benefits. But there is no relevant effectiveness analysis in this article, such as how much dental caries can be prevented after accepting fluoride varnish. It is recommended to add a paragraph explaining the current fluoride policy and related effects implemented in Australia.

Dental assistants are not equivalent to dental hygienists in some countries. DA have not received relevant knowledge and training, cannot conduct any business related to the oral cavity of patients under the medical laws. Therefore, this article seems insufficient just talk about cost alone. Is fluorine varnish allowed in the scope of job of a DA? Furthermore, it is recommended to explain how to ensure the quality of fluorine varnish between a DA and a OHT.

Line 187-190. The authors claimed that Mishra et al. found a range of one to four minutes per application.

Actually, Mishra et al. reported that the time required to apply the varnish varies from 1 to 4 min per preschool child, depending on the number of teeth present.

Fluorine varnish can indeed be completed within 2 minutes. If there is no oral examination and oral health education in this program, the fluoride application will lose the true meaning of prevention. The effectiveness of dental caries prevention will also be affected and cannot be fully utilized. Taiwan implements fluorine varnish twice a year, and the government stipulates that each doctor can only apply 30 children at a time (4 hours in morning or afternoon) to ensure quality.

Minor correction

Line 76-77. Please revise the sentence of “…(Index of Community 76 Socio-Educational Advantage) ICSEA rating of a 1050 or less” to ”… Index of Community 76 Socio-Educational Advantage (ICSEA) “.

Line 98. we used salary levels indicative of our experience to date in implementing pilot programs in NSW.. Please delete a “.”

Line 98. Please add a “.” of sentence after the Mishra et al.

Author Response

Point 1: However, the significance of prevention lies in the development of good tooth-brushing behavior in daily life. The weakness of this program background does not explain the current tooth-brushing behavior among of children, and the background, measures, and related results of domestic fluoride applications in Australian.

Response 1: The study is an investigation of the cost of scaling up a fluoride varnish program and is not concerned with toothbrushing. While some programs aimed at improving oral health among children include both fluoride varnish and toothbrushing, this is not the case with the program we are investigating. It is likely that the introduction of a toothbrushing program would have a substantial impact of the cost of scaling up a fluoride varnish intervention in the population that we are studying. We have also included more background in the Introduction that addresses some of the concerns raised (see Line 61-68).

Point 2: Materials and Methods

This article lacks the procedures for applying fluoride in the Materials and Methods section. Should fluoride be applied after teeth cleaning? Do you implement relevant oral hygiene education during the fluoride application process? It is recommended that the author should talk more about the oral hygiene status about these issues.

Response 2: A reference to the protocol and pilot study have now been added to the revised manuscript (see Lines 97-98). While a number of programs include oral health education, water bottles, fissure sealants, and/or toothbrushing we wanted to specifically cost the standard fluoride varnish component in keeping with the scale-up goals of various state and national plans.

Point 3: Result

Cost of Labour DA ($403) was much higher than Labour OHT ($206). Please check it $403 of Labour DA column in Table 4.

Response 3: Tables 4 and 5 have been revised in the amended manuscript.

Point 4: Discussion

Fluoride varnish is painted on the top and sides of each tooth with a small brush. The small brush is a consumable product, and the small brush is not mentioned in the material and cost estimation.

Response 4: The costs of the micro brushes has now been included in our costing and this is included in Table 3 (Line 161) and in the text of the amended manuscript (see Line 157).

Point 5: All health policies need to evaluate costs and benefits. But there is no relevant effectiveness analysis in this article, such as how much dental caries can be prevented after accepting fluoride varnish. It is recommended to add a paragraph explaining the current fluoride policy and related effects implemented in Australia.

Response 5: The study was not a cost effectiveness study. That would require a different kind of analysis and the collection of additional data which was not available to us. The cost of scale up however, is a legitimate avenue of enquiry as cost, rather than effectiveness, has been identified as a barrier to scale up in NSW. We have also included more background in the Introduction that addresses some of the concerns raised (see Line 61-68).

Point 6Dental assistants are not equivalent to dental hygienists in some countries. DA have not received relevant knowledge and training, cannot conduct any business related to the oral cavity of patients under the medical laws. Therefore, this article seems insufficient just talk about cost alone. Is fluorine varnish allowed in the scope of job of a DA? Furthermore, it is recommended to explain how to ensure the quality of fluorine varnish between a DA and a OHT.

Response 6: We have updated the introduction to explain the role of DAs in applying fluoride varnish here (see Lines 61-68). We have also added the need to explore quality differences between DAs and OHTs in the Limitations section (see Lines 263-264).

Point 7Line 187-190. The authors claimed that Mishra et al. found a range of one to four minutes per application.

Actually, Mishra et al. reported that the time required to apply the varnish varies from 1 to 4 min per preschool child, depending on the number of teeth present.

Response 7: This has been reworded, as suggested, in amended manuscript (see Lines 202-203).

Point 8Fluorine varnish can indeed be completed within 2 minutes. If there is no oral examination and oral health education in this program, the fluoride application will lose the true meaning of prevention. The effectiveness of dental caries prevention will also be affected and cannot be fully utilized. Taiwan implements fluorine varnish twice a year, and the government stipulates that each doctor can only apply 30 children at a time (4 hours in morning or afternoon) to ensure quality.

Response 8: Thank you for the comment. We have clarified that no oral health education is included in the costed program in the revised manuscript. We believe that the program is still effective without this component, but to be more conservative in our estimates have increased the time per application to three minutes. While a number of programs include oral health education, water bottles, fissure sealants, and/or toothbrushing we wanted to specifically cost the standard fluoride varnish component in keeping with the scale-up goals of various state and national plans.

Point 9: Line 76-77. Please revise the sentence of “…(Index of Community 76 Socio-Educational Advantage) ICSEA rating of a 1050 or less” to ”… Index of Community 76 Socio-Educational Advantage (ICSEA) “.

Response 9: Typographical error now corrected in amended manuscript (see Line 81).

Point 10: Line 98. we used salary levels indicative of our experience to date in implementing pilot programs in NSW.. Please delete a “.”

Response 10: Typographical error now corrected in amended manuscript (see Line 99).

Point 11: Line 98. Please add a “.” of sentence after the Mishra et al.

Response 11: Typographical error now corrected in amended manuscript (see Line 202).

Round 2

Reviewer 2 Report

Please see my remarks

Author Response

Point 1 Remove Full stop. Response: Full stop removed Line 4;

Point 2 Statistical model or method? Response: Word "method" added in Line 24;

Point 3 References. Response: References added in Line 48;

Point 4 Add comma. Response: Comma added Line 56;

Point 5 Add comma. Response: Comma added Line 100;

Point 6 Irrelevant data with..... Response: Sentence deleted (see Line 114);

Point 7 Add comma. Response: Comma added Line 135;

Point 8 Add comma. Response: Comma added Line 145;

Point 9 Line numbering issues. Fixed by Editorial Team (Line 182).

Reviewer 3 Report

I think the authors have made the best modification about his article. Please accept the article and I am optimistic about the outcome of this national primary school-based fluoride varnish program.

Author Response

Thank you for your comments and optimism.